# Cepharanthine Ameliorates Pulmonary Fibrosis by Inhibiting the NF-κB/NLRP3 Pathway, Fibroblast-to-Myofibroblast Transition and Inflammation

**DOI:** 10.3390/molecules28020753

**Published:** 2023-01-11

**Authors:** Guangrui Chen, Jian Li, Huimeng Liu, Huiyu Zhou, Mingqiu Liu, Di Liang, Zhiyun Meng, Hui Gan, Zhuona Wu, Xiaoxia Zhu, Peng Han, Taoyun Liu, Ruolan Gu, Shuchen Liu, Guifang Dou

**Affiliations:** Department of Pharmaceutical Sciences, Beijing Institute of Radiation Medicine, Beijing 100850, China

**Keywords:** cepharanthine, pulmonary fibrosis, myofibroblast, NF-κB/NLRP3, COVID-19

## Abstract

Pulmonary fibrosis (PF) is one of the sequelae of Corona Virus Disease 2019 (COVID-19), and currently, lung transplantation is the only viable treatment option. Hence, other effective treatments are urgently required. We investigated the therapeutic effects of an approved botanical drug, cepharanthine (CEP), in a cell culture model of transforming growth factor-β1 (TGF-β1) and bleomycin (BLM)-induced pulmonary fibrosis rat models both in vitro and in vivo. In this study, CEP and pirfenidone (PFD) suppressed BLM-induced lung tissue inflammation, proliferation of blue collagen fibers, and damage to lung structures in vivo. Furthermore, we also found increased collagen deposition marked by α-smooth muscle actin (α-SMA) and Collagen Type I Alpha 1 (COL1A1), which was significantly alleviated by the addition of PFD and CEP. Moreover, we elucidated the underlying mechanism of CEP against PF in vitro. Various assays confirmed that CEP reduced the viability and migration and promoted apoptosis of myofibroblasts. The expression levels of myofibroblast markers, including COL1A1, vimentin, α-SMA, and Matrix Metallopeptidase 2 (MMP2), were also suppressed by CEP. Simultaneously, CEP significantly suppressed the elevated Phospho-NF-κB p65 (p-p65)/NF-κB p65 (p65) ratio, NOD-like receptor thermal protein domain associated protein 3 (NLRP3) levels, and elevated inhibitor of NF-κB Alpha (IκBα) degradation and reversed the progression of PF. Hence, our study demonstrated that CEP prevented myofibroblast activation and treated BLM-induced pulmonary fibrosis in a dose-dependent manner by regulating nuclear factor kappa-B (NF-κB)/ NLRP3 signaling, thereby suggesting that CEP has potential clinical application in pulmonary fibrosis in the future.

## 1. Introduction

Pulmonary fibrosis (PF) is a chronic progressive and fatal fibrotic disease of the lungs with unclear etiology. Commonly, high-resolution electronic computed tomography (CT) of PF manifest as usual interstitial pneumonia (UIP). PF is characterized by fibroblast proliferation and excessive extracellular matrix (ECM) accumulation, accompanied by inflammatory damage and tissue structure destruction [1]. The origin of PF remains unknown in most cases, termed “idiopathic” [2]. Recently, the morbidity and mortality of PF have exhibited a gradual upward trend, with the median length of follow-up being as low as 3.5 years following diagnosis [3]. PF is more common in older adults over the age of 65 [4] and individuals with a history of smoking [5,6]. PF has also been established as sequelae of Corona Virus Disease 2019 (COVID-19) [7].

Over the past few decades, anti-inflammatory drugs, corticosteroids, and immunosuppressants have been used for treating PF according to the hypothesis that chronic inflammation in the alveolar space is the primary action mechanism of PF [8,9,10,11]. However, the above-mentioned treatments did not improve the survival or lung function of patients with PF [12]. Owing to progress in the understanding of the pathogenesis of PF, new drugs were developed, such as pirfenidone (PFD) and nintedanib, both of which are growth factor inhibitors approved by the Food and Drug Administration (FDA). PFD is the first drug to demonstrate some efficacy in PF in a replicated, randomized, placebo-controlled phase III clinical trial that reduced the decline of lung function in patients with PF [9] but did not improve survival or quality of life [10]. Currently, lung transplantation is the only viable therapeutic option known to improve PF prognosis [13], which requires further advancements. Especially following the COVID-19 pandemic, a deeper understanding of the pathogenesis of PF is necessary for developing novel treatment strategies.

Transforming growth factor-β1 (TGF-β1), a primary profibrogenic cytokine in the progression of PF [14,15], induces fibroblast differentiation to myofibroblast, thereby generating high amounts of collagen. Myofibroblasts are critical in the pathogenesis of PF as they secrete an enormous amount of extracellular matrix (ECM) proteins, causing concomitant loss of lung elasticity and function [16]. Myofibroblasts are rich in intracellular proteins, such as Vimentin and α-smooth muscle actin (α-SMA) [17], while producing ECM proteins, such as Collagen Type I Alpha 1 (COL1A1) [18], which are the hallmarks of pulmonary fibrosis. The chief sources of myofibroblasts include fibroblast-to-myofibroblast transition (FMT) and epithelial–mesenchymal transition (EMT) [19,20]. Therefore, inhibiting the myofibroblast viability is essential for detecting the therapeutic targets of PF.

Cepharanthine (CEP) is isolated from plants of the genus *Stephania* which is one of the biggest genera belonging to moonseed family Menispermaceae. CEP is a bisbenzylisoquinoline (BBIQ) alkaloid and is generally more abundant in tubers than in roots and is also found in the leaves and seeds of the plant [21]. This alkaloid is found in several *Stephania* species, principally *S. rotunda* Lour and *S. cephalantha* Hayata [22]; but also in *S. sasakii* Hayata, *S. suberosa* Forman, *S. epigaea* [23] and *S. japonica*. From a chemical perspective, CEP is also known as 12-O-methyl cepharanoline [24] and exhibits multiple pharmacological properties, including anti-inflammatory [25], immuno-regulatory [26], anticancer [27], and antiviral properties [28,29,30]. In a previous study, we found that CEP significantly reduced bleomycin (BLM)-induced collagen accumulation and inflammatory responses, engendering a protective effect against PF [31]. However, to the best of our knowledge, no studies have reported the underlying action mechanism of CEP in treatment against PF.

This study aimed to assess whether CEP exhibits anti-PF activity in animal models of PF and to determine the mechanisms underlying these effects. Here, we employed a BLM-induced PF rat model to explore the antifibrotic activity of CEP. In addition, human embryonic lung fibroblasts (MRC-5) cells were treated with TGF-β1 to evaluate the effect of CEP on the proliferation, migration, and differentiation of myofibroblasts. Furthermore, we provide the first evidence that CEP inhibits the nuclear factor kappa-B (NF-κB)/NOD-like receptor thermal protein domain associated protein 3 (NLRP3) signaling pathway in vitro, suggesting a possible mechanism by which CEP alleviates PF.

## 2. Results

### 2.1. CEP Attenuates BLM-Induced Collagen Deposition In Vivo

The rat model of BLM-induced PF has been widely used to explore the pathogenesis and potential novel therapies of PF [32]. PFD is one of two FDA-approved drugs for the treatment of PF, so it is often used as a positive control. Herein, after administering BLM to rats to induce PF, we treated them with CEP and PFD 24 h later for 21 consecutive days (Figure 1A). To investigate the effect of CEP on the lung tissue on a BLM-induced PF rat model, we examined the pathological changes in lung tissue and alterations in the levels of myofibroblast markers α-SMA and COL1A1 after 3 weeks. Compared with the control group, hematoxylin eosin (H&E) staining in the PF rats demonstrated that the alveolar septa and alveolar collapse had thickened (Green arrow) and H&E staining was accompanied by high granulocyte infiltration (Black arrow). Conversely, these symptoms were significantly relieved following 15 mg/kg CEP and 100 mg/kg PFD treatment, while 5 mg/kg CEP did not significantly improve BLM-induced lung inflammation (Figure 1B). Masson staining demonstrated proliferative deposition of blue collagen fibers in the lung of PF rats (Black arrow), which exhibited improvement after 15 mg/kg CEP and 100 mg/kg PFD treatment. However, 5 mg/kg CEP did not significantly improve BLM-induced lung collagen fiber generation in rats (Figure 1C). Next, we measured the α-SMA and COL1A1 expression using Western blotting, suggesting that 15 mg/kg CEP and 100 mg/kg PFD could effectively reduce the changes of protein levels, while 5 mg/mg CEP did not reduce the expression levels of these proteins (Figure 1D). Hence, these results suggest that 15 mg/kg CEP and 100 mg/kg PFD can ameliorate lung injury and collagen deposition in BLM-induced PF.

### 2.2. Effects of CEP on TGF-β1-Induced Apoptosis, Viability, and Migration of Myofibroblasts In Vitro

TGF-β1 plays a central role in PF by promoting FMT [33]. Embryonic lung fibroblasts (MRC-5) cells are commonly used as model cells to model pulmonary fibrosis cells [34]. Hence, to better understand the anti-PF mechanism of CEP, we induced MRC-5 transformation into myofibroblasts via TGF-β1. To assess whether CEP could alleviate PF by inhibiting the viability of myofibroblasts, we used cell counting kit-8 (CCK-8), Annexin V-FITC/PI, and cell wound-healing assays. We found that CEP inhibited cell viability and the inhibitory ability of the prophylactic administration was significantly higher than that of the therapeutic administration. Prophylactic administration of CEP had inhibitory effect on myofibroblasts at 0.2 μg/mL (*p* < 0.001), and the survival rate of myofibroblasts was unchanged when CEP ≥ 2 μg/mL. Therapeutic administration of CEP at 0.4 μg/mL had an inhibitory effect on myofibroblasts (*p* < 0.01), and the survival rate of myofibroblasts was unchanged when CEP ≥ 4 μg/mL. Hence, we used prophylactic CEP administration for the subsequent experiments, that concentration ranges from 0.2 μg/mL to 1 μg/mL (Figure 2A). Furthermore, we found that 10 ng/mL TGF-β1 could promote the migration of myofibroblasts compared to the control group, and all wounds healed within 48 h. However, CEP inhibited myofibroblast migration (Figure 2B). Image J was used to quantify the wound area between the two lines, and it was found that compared with only adding 10 ng/mL TGF-β1, 10 ng/mL TGF-β1 + CEP could inhibit wound healing, which meant that myofibroblast migration was inhibited, and the inhibiting capacity was proportional to the concentration of CEP (Figure 2C). Meanwhile, we detected apoptosis by Annexin V-FITC/PI. The sum of Q2 (Late apoptotic cell) and Q4 (Early apoptotic cell) cells was statistically analyzed, and we found that 10 ng/mL TGF-β1 did not affect the apoptosis of myofibroblasts compared with the control group. The intervention of CEP promoted the apoptosis of myofibroblasts, and the apoptosis rate was proportional to the concentration of CEP. The concentration higher than 0.8 ng/mL CEP had little effect on the proportion of early apoptotic cells, but only increased the proportion of late apoptotic cells (Figure 2D,E). Therefore, CEP inhibited FMT pathway by inhibiting myofibroblast cell migration and activity and promoting apoptosis, thereby reducing the degree of PF.

### 2.3. Mechanism of CEP for Regulating FMT in TGF-β1-Induced PF In Vitro

Our study showed that CEP inhibited TGF-β1-induced FMT; hence, we wanted to determine the regulatory role of CEP in TGF-β1-induced PF. Compared to cells from a control group, MRC-5 cells exposed to 10 ng/mL TGF-β1 expressed higher myofibroblast marker levels, including Fibronectin (Figure 3B), COL1A1 (Figure 3C), Vimentin (Figure 3D), Matrix Metallopeptidase 2 (MMP2) (Figure 3E), and α-SMA (Figure 3F), all of which are TGF-β1 profibrotic downstream effectors of the signaling pathway. However, Fibronectin, COL1A1 and Vimentin were reversed by CEP in a dose-dependent manner, except for 0.2 μg/mL CEP (Figure 3A–D), and their mRNA levels showed similar changes (Figure 3G,H,I), and α-SMA can be reversed by CEP in a dose-dependent manner (Figure 3A,F), and its mRNA level showed similar change (Figure 3K). It should be noted that MMP was also reversed in a dose-dependent manner except for 0.2 and 0.4 μg/mL CEP (Figure 3A,E), and its mRNA level showed similar change (Figure 3J). In general, however, CEP inhibits related effector factors in the FMT pathway. Therefore, CEP targets the FMT to inhibit fibroblast differentiation in vitro.

### 2.4. CEP Suppressed TGF-β1-Induced Inflammation via the NF-κB/NLRP3 Pathway In Vitro

Next, we confirmed the activation of the NF-κB/NLRP3 pathway in a TGF-β1-stimulated cell model. Western blotting revealed that Phospho-NF-κB p65 (p-p65)/NF-κB p65 (p65) and NLRP3 were upregulated by TGF-β1 while inhibitor of NF-κB Alpha (IκBα) was downregulated; CEP (0.4, 0.8, and 1 μg/mL) reversed these effects (Figure 4A,C,D). Consistently, RT-qPCR confirmed increase in NLRP3 mRNA levels due to TGF-β1 stimulation, and CEP (0.4, 0.8, and 1 μg/mL) inhibited these effects (Figure 4B); however, 0.2 μg/mL CEP did not regulate the above indexes, suggesting that the CEP (0.4, 0.8, and 1 μg/mL) inhibited the expression of NLRP3 in the inflammatory body in PF, and had anti-inflammatory effect, and inhibited the proliferation of myofibroblasts by inhibiting the activation of NF-κB by activating IκBα.

### 2.5. CEP Ameliorates PF by Inhibiting NF-κB/NLRP3-Induced FMT and Inflammation

In vitro studies, CEP decreased the expression of myofibroblast markers: Fibronectin, COL1A1, Vimentin, α-SMA, and MMP2 by inhibiting the FMT pathway. Moreover, by inhibiting the expression of NLRP3, the inflammatory level was reduced, while by increasing the expression of IκBα, the activation of NF-κB was inhibited. These combined effects made CEP reduce the PF level of MRC-5 cells (Figure 5).

## 3. Discussion

PF is a chronic progressive lung disease characterized by severe parenchymal scar formation, progressive ECM deposition [35], and increased myofibroblast formation, leading to the destruction of lung tissue structure and deterioration of lung function [15]. BLM treatment is known to increase inflammation and collagen deposition in lung tissue [36]. After 3 weeks, we detected lung tissue inflammation in the BLM-induced rat PF model, as indicated by H&E and Masson’s trichrome staining, as well as increased collagen fibers and damaged lung structure. Furthermore, we found increased collagen deposition, as indicated by the elevated expression levels of α-SMA and COL1A1, this is consistent with the results of previous studies [37], and which were significantly alleviated by PFD and CEP administration. PFD, the first FDA-approved treatment for pulmonary fibrosis, has been shown to slow the decline of lung function in PF patients [38], while CEP is comparable in efficacy.

Excessive myofibroblast formation causes persistent ECM deposition [39]. The cytokine TGF-β1, one of the most potent and well-studied inducers for FMT [40], creates a profibrotic environment by promoting fibroblast differentiation into myofibroblasts [41]. TGF-β1 activation also promotes cell migration and invasion under PF disease conditions [42]. Consistently, we observed that TGF-β1-induced FMT which promoted myofibroblast differentiation, proliferation, and migration. Nevertheless, CEP inhibited these activities. Therefore, we investigated the basal expression of fibrotic markers associated with the PF cascade. In TGF-β1-induced FMT, TGF-β1 increased the expression levels of fibronectin, COL1A1, vimentin, α-SMA, and MMP2, whereas CEP significantly inhibited the expression of the above indicators, thereby indicating that FMT was inhibited by CEP; however, the regulatory mechanism underlying these effects remains to be further explored.

Among BBIQ alkaloids, CEP is the only approved drug for human use in Japan since the 1950s and has been used to treat a number of acute and chronic diseases, including leukopenia, snake bites, xerostomia, and alopecia [24]. Recently, CEP was reported to exhibit strong antiviral activity against COVID-19 [28,29,43]. Concurrently, the anti-inflammatory property of CEP enables its use in the treatment of cerebral ischemia/reperfusion injury, wherein CEP inhibits NLRP3 signaling, and caspase-1, thereby suppressing the overproduction of IL-1β and IL-18, thus attenuating cerebral ischemia/reperfusion injury [44]. Moreover, CEP ameliorates chondrocytic inflammation and osteoarthritis by regulating the MAPK/NF-κB-autophagy pathway [45]. CEP also inhibited NF-κB activation, IκBα degradation, and ERK, JNK, and p38 phosphorylation in lipopolysaccharide-stimulated RAW264.7 cells in vitro and lipopolysaccharide-induced lung injury model in a dose-dependent manner [46]. Our results show that CEP inhibited IκBα degradation, NF-κB phosphorylation, and the PF cascade by suppressing the activation of the NF-κB pathway. Moreover, CEP inhibited the activation of the NLPP3 inflammasome, thereby reducing the inflammatory response and avoiding the aggravation of PF by circumventing the cytokine storm.

In conclusion, our study shows that CEP alleviated collagen deposition and inflammation in BLM-induced pulmonary fibrosis rats. Its potency is comparable to PFD, an FDA-approved drug for PF. Nevertheless, our animal model has certain limitations, as BLM models do not completely match all features of human disease [47]. However, this remains the best method for establishing animal models of PF disease [48]. Meanwhile, CEP inhibited fibroblast differentiation and reduced proliferation and migration of myofibroblasts by inhibiting FMT pathway. Moreover, CEP regulates the NF-κB/NLRP3 pathway and inhibits PF cascade events to mitigate the occurrence and progression of PF. However, the complex pathogenesis of PF necessitates further investigation into the molecular mechanisms underlying the potential therapeutic potential of CEP against PF caused by COVID-19.

## 4. Materials and Methods

### 4.1. Reagents and Materials

We used the following reagents: CEP (purity 99.7%, Topscience Co., Ltd., Shanghai, China), carbamazepine (purity 100%, National Institutes for Food and Drug Control, Beijing, China), BLM (purity 98.81%, MCE, Dallas, TX, USA), pirfenidone (purity 98%, Shanghai Macklin Biochemical Co., Ltd., Shanghai, China), and TGF-β1 (PEROTECH., Cranbury, NJ, USA).

### 4.2. Establishment of a PF Model and Animal Grouping

Male Sprague Dawley (SD) rats weighing 200 ± 20 g were purchased from Beijing Vital River Laboratory Animal Technology Co., Ltd. (license SCXK (Beijing) 2016-0011). The rats were accommodated for 7 days before experiments and housed at 50% ± 10% relative humidity, 24 °C ± 2 °C, and a 12 h light/dark cycle.

BLM is often used as an inducer to establish animal models of PF [49]. Animals were randomly divided into five groups: control, BLM (5 mg/kg)-induced PF, BLM (5 mg/kg)-induced PF with two different CEP concentrations (5 and 15 mg/kg), and BLM-induced PF with PFD (100 mg/kg) groups [50], each group had six rats. CEP was prepared with pH 3.7 acidic saline regulated by acetic acid, in which the CEP powder was fully dissolved by whirling for 20 s. The final pH value of the solutions was detected to be in the range of 5.0–5.5, which could be used for pulmonary administrations in the subsequent animal experiments. PFD was dissolved in 0.5% sodium carboxymethyl cellulose solution to prepare a suspension. All except the control rats were first anesthetized via isoflurane inhalation and administered with 5 mg/kg of BLM via spray atomization into the trachea to induce an inflammatory response and PF. Rats in the control group received spray atomization into the trachea of the same amount of saline solution as BLM. In the other groups where the pulmonary fibrosis model, rats were treated with CEP via spray atomization into the trachea or PFD via intragastric administration once per day for 21 days. After 21 days, rats were euthanized and their lung tissues were collected for further analysis. The assessment process was conducted by an assessor blind to the treatment allocation. All animal experiments were approved and supervised by the Institution of Animal Care and Use Committee, Academy of Military Medical Science (IACUC-AMMS, Beijing, China).

### 4.3. H&E Staining

HE staining is commonly used to observe structural changes and inflammatory reactions in tissues [51]. Isolated lung tissues were fixed using 4% polymerization at room temperature for 24 h, embedded into paraffin, and sliced into 4 μm sections using a rotary microtome. The dried slices were soaked in xylene, dewaxed for 10 min, and rehydrated; the hematoxylin stain was rinsed for 5 min and then washed, followed by differentiation with 1% hydrochloric acid alcohol. Next, slides were stained with eosin for 30 s, dehydrated using gradient alcohol, and soaked in xylene thrice. Finally, the slides were mounted with neutral gum and analyzed under an optical microscope (Eclipse Ci-L, Nikon, Japan). Image acquisition was conducted by an assessor blind to treatment allocation.

### 4.4. Masson’s Trichrome Staining

Masson’s trichrome staining is commonly used to detect collagen fiber formation in tissues [52]. The formation of collagen fibers was evaluated using Masson’s trichrome staining, per the manufacturer’s instructions. The glass sheet was dyed for 10 min in the working fluid of Weigert’s iron hematoxylin and washed in water to dye for 15 min in Masson Lichun red acidic compound red solution. After the color could be distinguished, the glass sheet was stained with aniline blue solution. Finally, the glass sheet was dehydrated with ethanol, clarified in dystershopine, the solid agent was sealed with resin, and analyzed under the optical microscope (Eclipse Ci-L, Nikon, Japan). Image acquisition was conducted by an assessor blind to treatment allocation.

### 4.5. Cell Culture and Transfection

MRC-5 was purchased from NMRC, Beijing, China. All cell lines in this study were authenticated by short tandem repeat (STR) profiling and analyzed for mycoplasma contamination. MRC-5 cells were grown in Minimum Essential Medium (MEM) (Gibco, New York, NY, USA) supplemented with 100 U/mL penicillin, 100 μg/mL streptomycin, 10% fetal bovine serum (HyClone, Logan, UT, USA), and Earle’s Balanced Salt Solution. Cells were cultured at 37 °C in a humid atmosphere with 5% CO_2_.

TGF-β1 is commonly used to induce the transformation of fibroblasts into myofibroblasts [53]. For cell treatment, the cells were divided into five groups: control group, TGF-β1 (10 ng/mL)-induced PF group and TGF-β1 (10 ng/mL)-induced PF with 0.2, 0.4, 0.8, or 1 μg/mL CEP groups. Cells were pretreated with 0.2, 0.4, 0.8, or 1 μg/mL of CEP for 1 h, then 10 ng/mL of TGF-β1 was added to induce PF for 48 h.

### 4.6. Determination of Cell Viability and Apoptosis

Cell viability was measured using CCK-8 [52,54] (CCK-8, Dojindo, Japan), per the manufacturer’s protocols. Briefly, MRC-5 cells were seeded in 96-well plates (1.5 × 10^4^) for 24 h. Then, the cells were serum starved in serum-free medium for 24 h. Next, the Prophylactic administration cells were incubated in MEM containing different CEP concentrations (0, 0.1, 0.2, 0.4, 0.8, 1, 2, 4, 8, and 16 μg/mL) for 1 h. Cells were subsequently exposed to 10 ng/mL TGF-β1 with or without CEP for 48 h. In contrast, for therapeutic administration, cells were exposed to 10 ng/mL TGF-β1 for 1 h, followed by the same CEP as prophylactic administration, and continued to be cultured for 48 h. Next, the media was replaced with MEM supplemented with 10% CCK-8 reagent, 100 μL of basal medium (MEM supplemented with 100 U/mL penicillin, 100 μg/mL streptomycin and Earle’s Balanced Salt Solution) mixed with CCK8 dye was added to the wells (CCK8 dye: MEM basal medium = 1:10), and the cells were cultured for another 2 h at 37 °C. Finally, the optical density was measured using microplate reader (BIO-TEK EIK800, Winooski, VT, USA) at 450 nm. Image acquisition was conducted by an assessor blind to treatment allocation.

Annexin V is a phospholipid binding protein that is one of the sensitive indicators for the detection of early apoptosis of cells. Propidium Iodide (PI) is a nucleic acid dye that causes the cell nucleus to become red by damaging cell membranes. Thus, Annexin V with PI can be used to distinguish cells at different stages of apoptosis [55]. MRC-5 cells at logarithmic growth phase were seeded in 6-well plates (1 × 10^5^). At 80% confluence, the cells were serum starved for 24 h in serum-free media; thereafter, the cells were grouped and treated as indicated. After 48 h, the culture media was discarded, the cells were collected via trypsin digestion after washing with precooled PBS, centrifuged, washed twice with precooled PBS, counted, following the Annexin V–FITC/PI Apoptosis Detection Kit (BD, New York, NJ, USA) instructions, resuspended with 1 × Annexin V binding buffer, and 100 μL of cell suspension (approximately 1 × 10^6^) was used for flow cytometry. Then, 5 μL FITC Annex and 5 μL FITC Annex PI were gently mixed and incubated at room temperature in the dark for 15 min. Afterward, 400 μL 1 × Annexin V binding buffer was added to each flow tube. The test was completed within 1 h. The test was performed by researchers blind to group allocation. Image acquisition was conducted by an assessor blind to treatment allocation.

### 4.7. Wound-Healing Assay

Wound-healing experiments are used to observe changes in tumor and fibroblast migration ability [56]. An equal number of MRC-5 cells were seeded in 6-well plates. At 100% confluence, the cells were serum starved for 24 h in serum-free media; thereafter, two scratch wounds were made using a sterile P-200 pipette-tip in each well, which was washed thrice with precooled PBS to remove the scratched cells. Then, following the group instructions, the cells were treated with CEP for 1 h. Following 1 h incubation, TGF-β1 (10 ng/mL) was added to all the wells except the vehicle control well. Photomicrographs were captured (Leica DMI3000B, Wetzlar, HESSE, Germany) at 0, 24, and 48 h post-wound generation. Image acquisition was conducted by an assessor blind to treatment allocation. Wound area was calculated using Image J software.

### 4.8. RNA Extraction and RT-qPCR

RT-qPCR is commonly used to observe the changes of relevant indicators at the transcription level [57]. Total RNA was isolated from frozen lung specimens and cells using TRIZOL Reagent (Thermo, Waltham, MA, USA) in accordance with manufacturer’s protocols. PrimeScript RT reagent kit (Vazyme Biotech Co., Ltd., Nanjing, China) was used to reverse-transcribe 1 ug RNA to complementary DNA. Gene expression was measured using RT-qPCR using an instrument. The relative expression of PCR products was determined according to the 2−ΔΔCt method, which compares the target gene and GAPDH, used as the endogenous control. The following primer sequences were used: Hu-GAPDH-F, 5′-GGAGCGAGATCCCTCCAAAAT-3′; Hu-GAPDH-R, 3′-GGCTGTTGTCATACTTCTCATGG-5′; Hu-αSMA-F, 5′-GTGGATCACCAAGCAGGAGT-3′; Hu-αSMA-R, 3′-TTCGTCGTCCTGAGAAGTCG-5′; Hu-COL1A1-F, 5′-GCCTCTGCTCTCCGACCTCTC-3′; Hu-COL1A1-R, 3′-CTGCTTTGTGCTTTGGGAAGTTGTC-5′; Hu-NLPR3-F, 5′-CTTGCCGACGATGCCTTCCTG-3′; Hu-NLPR3-R, 3′-GCTGTCATTGTCCTGGTGTCTTCC-5′; Hu-Fibronectin-F, 5′-TCAGCTTCCTGGCACTTCTG-3′; Hu-Fibronectin-R, 3′-TCTTGTCCTACATTCGGCGG-5′; Hu-Vimentin-F, 5′-GGACCAGCTAACCAACGACA-3′; Hu-Vimentin-R, 3′-AAGGTCAAGACGTGCCAGAG-5′; Hu-MMP2-F, 5′-TGATGGCATCGCTCAGATCC-3′; Hu-MMP2-R, 3′-GGCCTCGTATACCGCATCAA-5′.

### 4.9. Western Blotting

Western blotting is used to measure the change of related indicators in protein expression level [58]. Proteins were extracted from cells or tissues using RIPA lysates. Total protein concentration was measured using BCA protein assay kit (Thermo, Waltham, MA, USA). The same amount of protein was loaded in each lane of 10% sodium dodecyl sulfate–polyacrylamide gel electrophoresis gels, followed by electrophoresis and protein transfer to polyvinylidene fluoride membranes (Millipore, Boston, MA, USA). Following the transfer, the membranes were blocked with 5% skim milk. Immunoblots were probed with primary antibodies against vimentin (1:1000, Cell Signaling Technology, Danvers, MA, USA), COL 1A1 (1:1000, Cell Signaling Technology, Danvers, MA, USA), α-SMA (1:1000, Cell Signaling Technology, Danvers, MA, USA), fibronectin (1:5000, Abcam, Cambridge, UK), MMP-2 (1:2000, Abcam, Cambridge, UK), NLRP3 (1:1000, Abcam, Cambridge, UK), p-p65 (1:1000, Cell Signaling Technology, Danvers, MA, USA), p65 (1:1000, Cell Signaling Technology, Danvers, MA, USA), IκBα (1:5000, Abcam, Cambridge, UK), or GAPDH (1:1000, Cell Signaling Technology, Danvers, MA, USA) at 4 °C overnight followed by goat antirabbit horseradish peroxidase-labeled secondary antibodies (1:2000, Cell Signaling Technology, Danvers, MA, USA) for 1 h at room temperature. After extensive washing, the immunoblots were visualized using ECL (APPLYGEN, Beijing, China). Then, band intensity was quantified using ImageJ software.

### 4.10. Statistical Analysis

All experiments were independently repeated in triplicate. Data were analyzed using Prism 8.3.0, GraphPad Software. All data were expressed as mean ± SD. Comparison of differences among groups was performed using one-way analysis of variance. *p* < 0.05 was considered statistically significant.

## Figures and Tables

**Figure 1 molecules-28-00753-f001:**
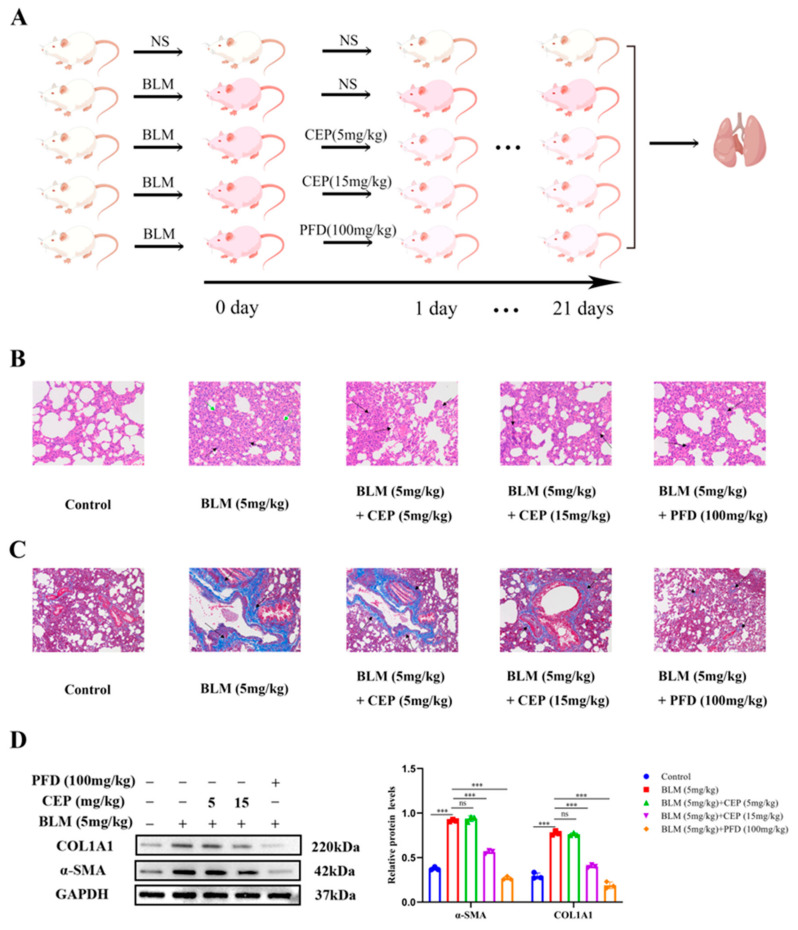
CEP attenuates BLM-induced collagen deposition in rats. (**A**): Experimental process in vivo (By Figdraw), *n* = 6. (**B**): Lung tissue structures inflammatory expression visualized using hematoxylin and eosin (H&E) staining. alveolar septa and alveolar collapse had thickened (Green arrow). high granulocyte infiltration (Black arrow). Scale bar: 100 μm. (**C**): Collagen deposition in lung tissues visualized using Masson’s trichrome staining. blue collagen fibers (Black arrow). Scale bar: 100 μm. (**D**): Protein levels of α-SMA and COL1A1 measured with Western blotting. ns: no statistical difference, and *** *p* < 0.001.

**Figure 2 molecules-28-00753-f002:**
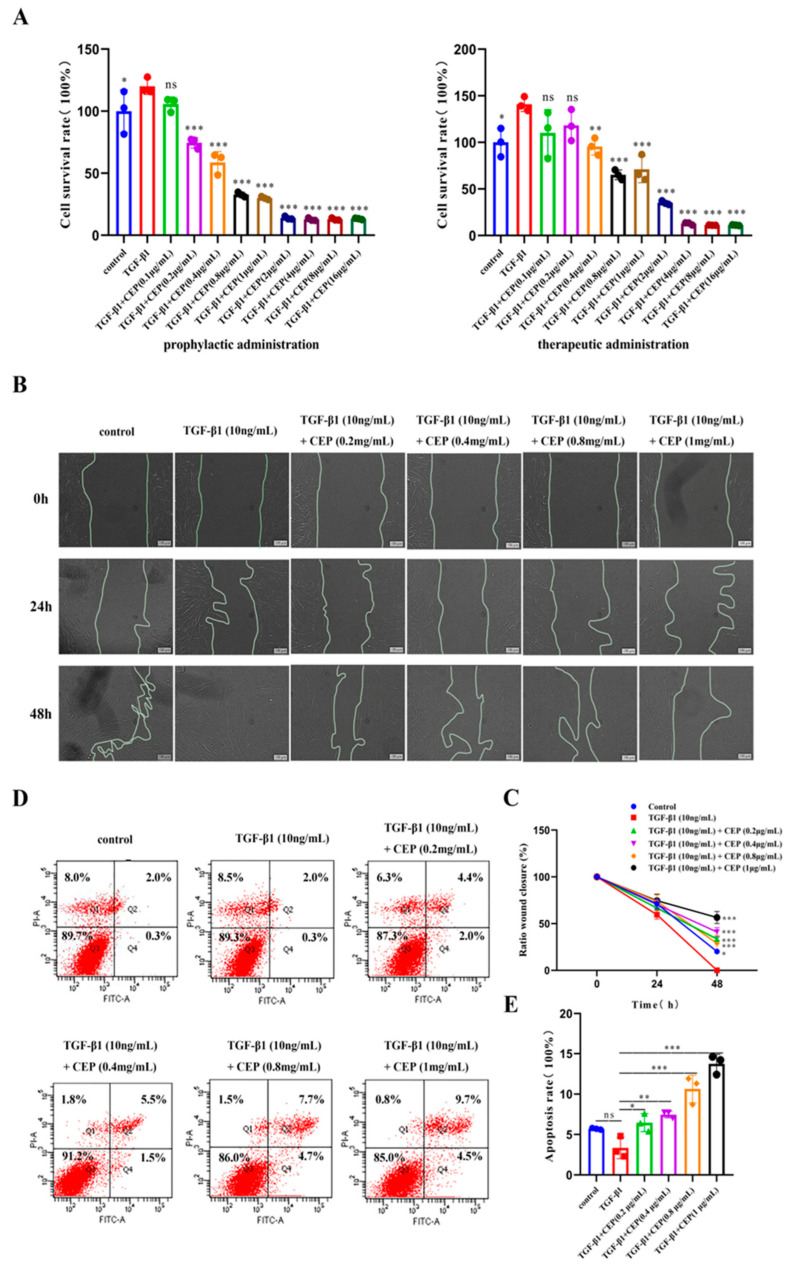
Effects of CEP on TGF-β1-induced apoptosis, viability, and migration of myofibroblasts in vitro. (**A**): Effect of prophylactic and therapeutic CEP administration on the survival rate of myofibroblasts as investigated with CCK-8 assay (“prophylactic administration” is to add different concentrations of CEP first, without removing CEP, then add 10 ng/mL TGF-β1 1 h later, and continue the culture 48 h; “therapeutic administration” is to add 10 ng/mL TGF-β1 for 1 h, without removing TGF-β1 for 1 h and then add different concentrations of CEP, and continue the culture 48 h.). (**B**): Effect of CEP on myofibroblast migration with wound-healing assay. (**C**): Image J software was used to quantify the area of wound (the area between two lines). (**D**): Effect of CEP on myofibroblast apoptosis with Annexin V-FITC/PI. Q1: Necrotic cell; Q2: Late apoptotic cell; Q3: Living cell; Q4: Early apoptotic cell. (**E**): The sum of the percentage of Q2 and Q4 regions of each group was analyzed. ns: no statistical difference, * *p* < 0.05, ** *p* < 0.01, and *** *p* < 0.001.

**Figure 3 molecules-28-00753-f003:**
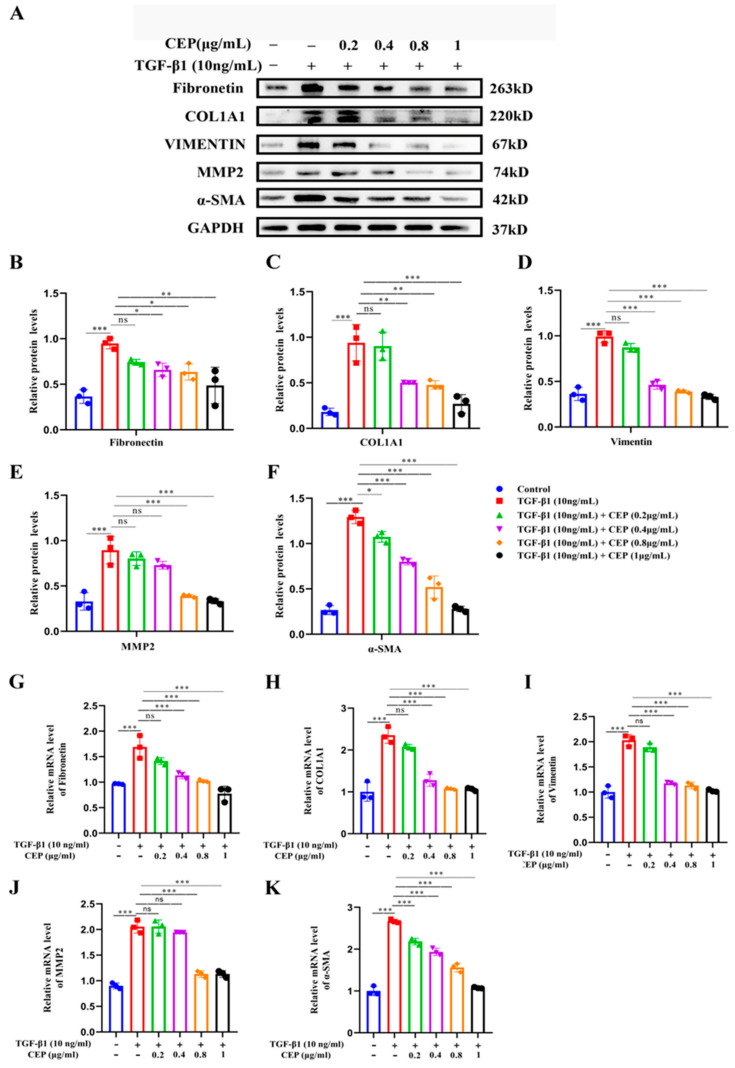
Mechanism of CEP for regulating FMT in TGF-β1-induced PF in vitro. (**A**): Protein levels of Fibronectin, COL1A1, Vimentin, MMP-2, and α-SMA measured with Western blotting, respectively. (**B**): Fibronectin protein level was quantified by grayscale value and Image J software was used. (**C**): COL1A1 protein level was quantified by grayscale value and Image J software was used. (**D**): Vimentin protein level was quantified by grayscale value and Image J software was used. (**E**): MMP-2 protein level was quantified by grayscale value and Image J software was used. (**F**): α-SMA protein level was quantified by grayscale value and Image J software was used. (**G**): The mRNA levels of fibronectin measured with RT-qPCR. (**H**): The mRNA levels of COL1A1 measured with RT-qPCR. (**I**): The mRNA levels of vimentin measured with RT-qPCR. (**J**): The mRNA levels of MMP-2 measured with RT-qPCR. (**K**): The mRNA levels of α-SMA measured with RT-qPCR. ns: no statistical difference, * *p* < 0.05, ** *p* < 0.01, and *** *p* < 0.001.

**Figure 4 molecules-28-00753-f004:**
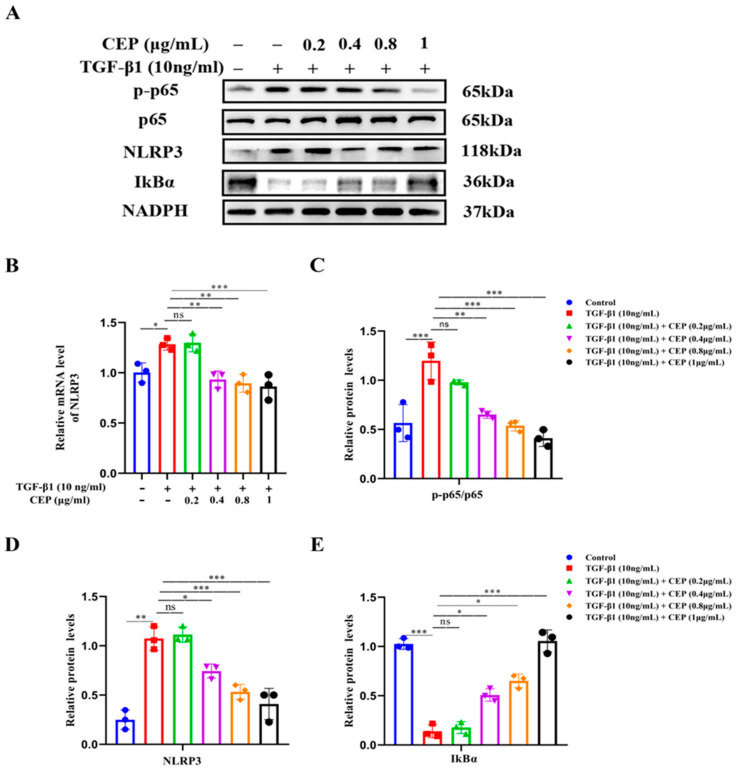
CEP suppressed TGF-β1-induced inflammation via the NF-κB/NLRP3 pathway in vitro. (**A**): Protein levels of NLRP3, phosphorylated p65 (p-p65), p65, and IκBα measured with Western blotting, respectively. (**B**): mRNA levels of NLRP3 measured with RT-qPCR. (**C**): The ratio of the amount of protein level in p-p65/p65 with gray value and Image J software was used. (**D**): NLRP3 protein level was quantified by grayscale value and Image J software was used. (**E**): IκBα protein level was quantified by grayscale value and Image J software was used. ns: no statistical difference, * *p* < 0.05, ** *p* < 0.01, and *** *p* < 0.001.

**Figure 5 molecules-28-00753-f005:**
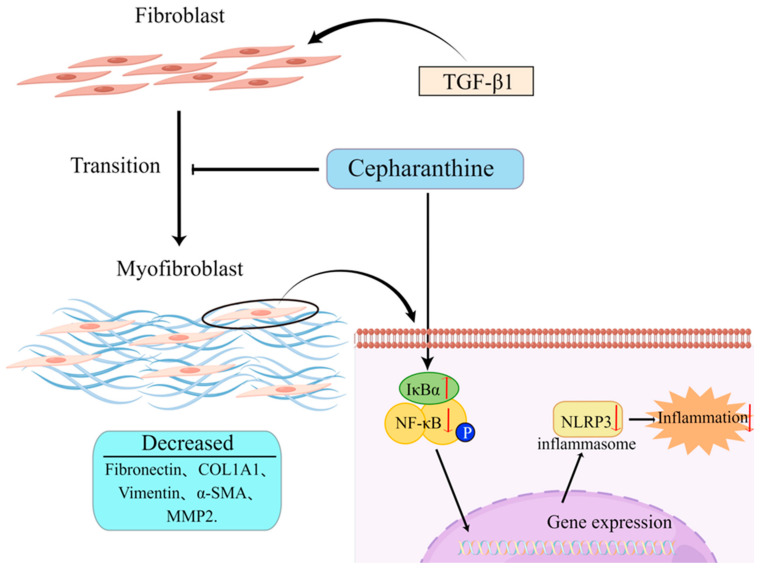
Schematic of the hypothesis of this study (by Figdraw). CEP suppressed inflammation and FMT in TGF-β1-induced PF via the NF-κB/NLRP3 signaling pathway.

## Data Availability

The data presented in this study are available on request from the corresponding author.

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
