# Peer review of "Cepharanthine Ameliorates Pulmonary Fibrosis by Inhibiting the NF-κB/NLRP3 Pathway, Fibroblast-to-Myofibroblast Transition and Inflammation"

_molecules, 2023, doi:10.3390/molecules28020753_

Round 1
Reviewer 1 Report
The manuscript about putative function of Cepharanthine in pulmonary fibrosis was in general well writen. Some minor issues need tobe improved.
1. Fig1D: please also use dot plot to present the qPCR results as other figure in this manuscript.
2. line 160: is here a full paragraph (2.5) missing? or it is just a duplicated subtitle of 2.4.
3. Please discuss more about pirfenidone (PFD). Why pfd was used as a positive control in in vivo study, but not used in any in vitro assays.
Author Response
Response to Reviewer 1 Comments
Point 1. Fig1D: please also use dot plot to present the qPCR results as other figure in this manuscript.
Response 1. Thanks for your feedback. 1D plots have been corrected.
2. line 160: is here a full paragraph (2.5) missing? or it is just a duplicated subtitle of 2.4.
Response 2. Paragraph (2.5) is a summary figure summarizing the results of all in vitro experiments. The results in the figure are described in detail in each in vitro experiment, so paragraph (2.5) will not repeat the text description. A short description has been added.
3. Please discuss more about pirfenidone (PFD). Why pfd was used as a positive control in in vivo study, but not used in any in vitro assays.
Response 3. Pirfenidone (PFD). has been added to the discussion. PFD has been used as a positive control in vivo studies because it is an FDA-approved drug for the treatment of pulmonary fibrosis. In vivo experiments have shown that cefaxanthin has an inhibitory effect on pulmonary fibrosis in rats. In the in vitro test, I mainly wanted to study the inhibitory mechanism of cefaxanthin on the pulmonary fibrosis cell model, not its efficacy, so I did not choose a positive control.
Reviewer 2 Report
The authors studied the impact of a molecule name: Cepharanthine (CEP) on pulmonary fibrosis. They show that this molecule can ameliorate the state of inflammation by inhibition of myofibroblast migration and apoptosis. They have studied the impact of this molecule on the RNA and protein level of a range of proteins participationg in fibrosis loading, as Collagen , alpha sma , MMP 2, vimentin and fibronectin.
They also used animal models for testing their molecule on lung tissue directly.
Comments:
1- Why they choose inhalation instead of oral administration?
2- Why authors did not present the result of blind evaluation of fibrotic stat on animal model. Normally pathologist or here they speak about blind assessor, perform some quantification analysis on amount of fibrotic tissue on the slides to categorise or annoce the stage or degree of fibrosis. Even researcher can performe the statistical analysis on their quantifications. I propose the authors to present these tables and add the evaluation table of pathologist in material and method section, means how he or she evaluate the degree of fibrosis on the pulmonary tissue. Then , they need to present these results in results section and they can add some discussion about this modeling in there discussion section.
3-It would be really appreciated if they can explain about their choice on animal model, which consists of only one stimulation in a disease which is a chronic one, means continuously stimulated.
4- In protein expression level study, why they choose collagen 1 and MMP2, however MMP2 play a role in gelatine not collagen loading or degradation.
Author Response
Response to Reviewer 2 Comments
Point 1: Why they choose inhalation instead of oral administration?
Response 1: In previous studies, we found that cephanthine (CEP) is more bioavailable when inhaled and administered directly to the lungs compared to oral administration. Specific can reference DOI: 10.3390/molecules27092745
Point 2: Why authors did not present the result of blind evaluation of fibrotic stat on animal model. Normally pathologist or here they speak about blind assessor, perform some quantification analysis on amount of fibrotic tissue on the slides to categorise or annoce the stage or degree of fibrosis. Even researcher can performe the statistical analysis on their quantifications. I propose the authors to present these tables and add the evaluation table of pathologist in material and method section, means how he or she evaluate the degree of fibrosis on the pulmonary tissue. Then , they need to present these results in results section and they can add some discussion about this modeling in there discussion section.
Response 2: In the Masson staining experiment for the observation of blue collagen fibers, the pathological map showed obvious changes in the area of blue collagen fibers among all groups, so no quantitative analysis was carried out.
Point 3: It would be really appreciated if they can explain about their choice on animal model, which consists of only one stimulation in a disease which is a chronic one, means continuously stimulated.
Response 3: We chose to give a single dose of 5 mg/kg bleomycin to establish a rat model of pulmonary fibrosis because this method has been very mature, many literatures can be used as reference, and the success rate is high.
Point 3: In protein expression level study, why they choose collagen 1 and MMP2, however MMP2 play a role in gelatine not collagen loading or degradation.
Response 3: COL1A1 is one of the markers of fibrosis. MMP2 can not only degrade extracellular matrix proteins, but also inhibit cell migration and induce the activation signal of NF-κB. So I chose COL1A1 and MMP2.
Reviewer 3 Report
Pulmonary fibrosis (PF) is one of challenging diseases for humans. This article showed the potential use of catharanthine (CEP) as a combination therapy with bleomycin (BLM). This article seems to be the continuation of their efforts in the previous publications. (Molecules 2022, 27, doi:10.3390/molecules27092745.)
In this article, the authors tried to identify the mechanism of action, and the methods were well planned: The antifibrotic effects of CEP are by modulating the NF-κB/NLRP3 pathway and inhibiting the PF cascade events. Since the co-administration of CEP and BLM are unclear, it still needs to investigate these effects. However, for now, to find new therapeutics, the identification of targets is essential. The results of this study will be valuable in developing new therapeutic agents. I recommend that this manuscript to publish in Molecules as is.
Author Response
Response to Reviewer 3 Comments
Thank you very much for your recognition.
Reviewer 4 Report
Journal: MDPI-Molecules
Manuscript ID: molecules-2005610
Title: Cepharanthine ameliorates pulmonary fibrosis by inhibiting the NF-κB/NLRP3 pathway, fibroblast-to-myofibroblast transition and inflammation
Type of manuscript: Article
Comments
1.Abbreviation in title/subtitle/abstract must be avoided.
Abstract: Please write full form of “COL1A1, α-SMA, and MMP2”, NF-κB/NLRP3, p65, IκBα etc.
Introduction:
1. In this study, rats were treated with pirfenidone (PFD) along with Cepharanthine (CEP). Author has mentioned importance of CEP . However, why PFD treatment was given, not mentioned. The information of FDA approved PFD, its importance and why it is considered to be studied along with CEP etc must be incorporated.
2. Please provide detail information about Stephania cephalantha Hayata (source plant of CEP). Is CEP is the only alkaloid available from Stephania cephalantha Hayata. What are the other compounds that are reported to be used in various disease. Is CEP also available from leaves, stem and other parts of plants apart from tuber and roots? Information must be incorporated.
3. Line 39, it is written “…..survival rate being as low as 2.5–3.5 years following diagnosis”. Why survival rate is mentioned in years? Confusion? Please clarify.
4. Why human embryonic lung fibroblasts (MRC-5) cells were used in this study. What is its advantage. Why particularly TGF-β1 was used to treat the cells? Information must be incorporated along with references.
5. Line 40, it is written “The mortality rate of PF is higher than most tumors, and hence, it is known as a “tumor-like disease” and PF is more common in older adults”, here please clarify what are the other tumors, how much higher mortality rate and what is the age limit of older adults having PF.
6. Please incorporate mechanism of α-SMA and COL1A1 in PF pathophysiology .
Results:
1. Results for the measurement of α-SMA and COL1A1 expression by RT–qPCR (Fig 1D) and western is not sufficiently analysed. For ex: In Fig 1D, the expression level of α-SMA and COL1A1 were shown in which qRT-PCR data shows less expression at CEP dose (15 mg/kg) than PFD (100mg/ml) whereas Western data (Fig 1E), protein level data shows more expression at CEP dose (15 mg/kg) than PFD (100mg/ml). Please clarify and incorporate information.
2. Similarly Western results as shown in Fig 1E has not been analysed appropriately. BLM+CEP 5mg/kg shows upregulation of α-SMA and COL1A1 expression and is shown to be equivalent to BLM 5mg/kg. Please incorporate the analysis appropriately.
3. Line 110, It is written “we induced MRC-5 transformation into myofibroblasts via TGF-β1, and the transformation was marked by high α-SMA expression”, however relevant data is missing. Please incorporate relevant Figs.
4. Line 114, It is written “ …. prophylactic administration was significantly higher than that of the therapeutic administration”, here please clarify “prophylactic administration” and “therapeutic administration”. How the doses of prophylactic and therapeutic administration were decided. Information of “prophylactic administration” and “therapeutic administration” must be included.
5. Please provide data for induction of PF using different concentration TGF-β1
6. Results of Fig 2B must be elaborated so as to reach appropriate conclusion.
7 Results of Fig 3 needed to be analyzed and written elaborately followed by conclusion and discussion of the results .
For ex: Fig 3B, Vimentin and MMP2 expression is
higher than TGF-β1 (10ng/ml) at TGF-β1 (10ng/ml)+CEP (0.2ug/ml). Likewise other results must be analyzed and incorporated appropriately.
Methods:
1.There is no reference for the method used to do the said study. Please incorporate references for PF model generation, H&E staining, Masson’s trichrome staining, Cell culture and transfection, cell viability determination and apoptosis, Wound-healing assay, RNA extraction and RT–qPCR, Western blotting etc.
2. Animal Ethics approval reference number must be provided.
3.How author selected two different CEP concentrations 5 and 15 mg/kg, and PFD (100 mg/kg) to treat BLM-induced rat. How specifically these doses were selected? What are the criteria to select treatment time and dose of CEP and PFD and in which solvent CEP was dissolved. Please incorporate information along with reference.
4. Line 259, it is written “For cell treatment, the cells were divided into five groups: control group, TGF-β1 induced PF group, and TGF-β1-induced PF with 0.2, 0.4, 0.8, or 1 μg/mL CEP groups.” Here how do the author came to this concentration of CEP?, Please clarify along with references
5.Why cells were induced with 10 ng/mL of TGF-β1? Mention reference.
6.Why cells were pretreated with TGF-β1? And why CEP treatment was given for 1 hour while with TGF-β1 cells were incubated for 48 hours. Till this time CEP effect might be nullified? Please Justify.
7. Also how BLM concentration decided to be 5mg/kg to induce PF.
8.How many animals were included in each group of animals. Author made 5 groups of animals but standard drug control group was not included. What about the vehicle and sham control group (not sprayed in the trachea by saline). It must have been included for appropriate conclusion of results. Author included one animal group to which PFD treatment was given. What is the importance of PFD. Why PFD treatment was given along with CEP. Also why PFD and CEP treatment concentration was different. Please clarify and incorporate analysis and justify along with reference.
9.Why specifically two markers α-SMA and COL1A1 were considered to be studied in this study. Please provide importance of these two markers (α-SMA and COL1A1) in the present study.
10. Why different staining like H&E, Masson’s trichrome staining was carried out. What is its importance. Please provide information along with reference.
11. In 4.6, author mentioned in line 272 “ ……..basal medium mixed with CCK8 dye……”. Please provide details of basal medium.
12.Results of Fig 4(A-D) needed to be described carefully and elaborately.
For western it is written in line 150 “Western blotting revealed that p-p65/p65 and NLRP3 were upregulated by TGF-β1 while IκBα was downregulated”. However, careful analysis shows, p-p65/p65 expression little higher only at CEP conc 0.4 ug/ml (Fig 4c) than control whereas in other conc of CEP, the expression level is mostly equivalent to control. Similarly NLRP3 shows upregulated expression than control but at CEP conc 0.2 ug/ml NLRP3 shows more expression TGF-β1 (Fig 4D). Also IκBα expression was increased with increasing conc of CEP (Fig 4A).
Similarly results of Fig 3 also must be written elaborately. In Fig 3B, protein expression of vimentine and MMP-2 is little higher than TGF-β1at CEP conc 0.2 ug/ml. Results must be written attentively with appropriate justification.
13.Raw data of all western blot must be provided.
Figs/tables
1) Please incorporate sufficient details of all the Figs in the figure legend.
2) Full form of all abbreviation must be incorporated as a foot note.
3) Alveolar septa, alveolar collapse, alveolar cavity and high granulocyte infiltration in H&E staining in Fig 1B must be labelled. Similarly, proliferative deposition must be labelled in masson staining.
4) In Fig 1E, BLM concentration (5mg/kg) is not written in densitometric analysis. Please incorporate.
5) In wound healing assay photomicrographs were captured by using which instrument. Also in 4.6 author used which company ultraviolet spectrophotometer. Please add information
Discussion:
Discussion needed to be improved. It is not sufficiently strong.
Conclusion:
CEP is already known as anti-inflammatory agent. In this study the obtained results were not analyzed appropriately. Some results were written very short thus led to inappropriate conclusion. The mechanism that CEP alleviates the occurrence and progression of PF is not sufficiently justified. Also conclusion portion is missing. The authors did not optimized the CEP concentration and thus the best CEP concentration was not reported.
Others
1.Line 47, write correctly “abovementioned”
2. Line 180, Correct the sentence “Consistently, We observed that TGF-β1–induced FMT which myofibroblast differentiation, proliferation, and migration”.
3.Please incorporate full form of all abbreviation (FITC/PI, NMRC, PBS, RIPA,BCA, ECL, CST, ERK, JNK, NLPP3 etc). Full form of all abbreviation must appear when appear 1st time in the manuscript.

Author Response
Response to Reviewer 4 Comments
Point 1. Abbreviation in title/subtitle/abstract must be avoided.
Abstract: Please write full form of “COL1A1, α-SMA, and MMP2”, NF-κB/NLRP3, p65, IκBα etc.
Response 1. A full form of the relevant abbreviations has been added.
Introduction:
Point 2. In this study, rats were treated with pirfenidone (PFD) along with Cepharanthine (CEP). Author has mentioned importance of CEP. However, why PFD treatment was given, not mentioned. The information of FDA approved PFD, its importance and why it is considered to be studied along with CEP etc must be incorporated.
Response 2. Line 53 describes PFD-approved pirfenidone for pulmonary fibrosis and adds to the literature on its importance. Therefore, it can be used as a positive drug to compare with the efficacy of CEP.
Point 3. Please provide detail information about Stephania cephalantha Hayata (source plant of CEP). Is CEP is the only alkaloid available from Stephania cephalantha Hayata. What are the other compounds that are reported to be used in various disease. Is CEP also available from leaves, stem and other parts of plants apart from tuber and roots? Information must be incorporated.
Response 3. The relevant content has been added at line 72.
Point 4. Line 39, it is written “…survival rate being as low as 2.5–3.5 years following diagnosis”. Why survival rate is mentioned in years? Confusion? Please clarify.
Response 4. Description changed in line 42.
Point 5. Why human embryonic lung fibroblasts (MRC-5) cells were used in this study. What is its advantage. Why particularly TGF-β1 was used to treat the cells? Information must be incorporated along with references.
Response 5. TGF-β1 treatment of human embryonic lung fibroblast (MRC-5) cells to establish pulmonary fibrosis cell model is a common method to establish pulmonary fibrosis cell model. References have been added to line 123.
Point 6. Line 40, it is written “The mortality rate of PF is higher than most tumors, and hence, it is known as a “tumor-like disease” and PF is more common in older adults”, here please clarify what are the other tumors, how much higher mortality rate and what is the age limit of older adults having PF.
Response 6. In line 43, the description "PF has a higher mortality rate than most tumors, so it is referred to as" tumor-like disease "was deleted because no relevant literature was found. The age limit for PF has been added.
Point 7. Please incorporate mechanism of α-SMA and COL1A1 in PF pathophysiology.
Response 7. In line 65, The mechanism of α-SMA and COL1A1 in the pathophysiology of PF has been added in line 61 and references have been added.
Results:
Point 8. Results for the measurement of α-SMA and COL1A1 expression by RT–qPCR (Fig 1D) and western is not sufficiently analysed. For ex: In Fig 1D, the expression level of α-SMA and COL1A1 were shown in which qRT-PCR data shows less expression at CEP dose (15 mg/kg) than PFD (100mg/ml) whereas Western data (Fig 1E), protein level data shows more expression at CEP dose (15 mg/kg) than PFD (100mg/ml). Please clarify and incorporate information.
Response 8. In FIG. 1D and FIG. 1E, CEP (15mg/kg) and PFD (100mg/ml) showed inconsistent trends in western blotting and qRT-PCR expression, but there was no statistically significant difference between the two groups. Therefore, it can be considered that the two groups had similar effects on the expression levels of α-SMA and COL1A1 in pulmonary fibrosis rats.
Point 9. Similarly Western results as shown in Fig 1E has not been analysed appropriately. BLM+CEP 5mg/kg shows upregulation of α-SMA and COL1A1 expression and is shown to be equivalent to BLM 5mg/kg. Please incorporate the analysis appropriately.
Response 9. Western blot is qualitative and its gray value analysis can be used as reference. qRT-PCR was quantitative and its results were statistically significant.
Point 10. Line 110, It is written “we induced MRC-5 transformation into myofibroblasts via TGF-β1, and the transformation was marked by high α-SMA expression”, however relevant data is missing. Please incorporate relevant Figs.
Response 10. The results of high α-SMA expression are shown in Figure 3. Therefore, the statement "the transformation is marked by high α-SMA expression" in line 126 was deleted.
Point 11. Line 114, It is written “prophylactic administration was significantly higher than that of the therapeutic administration”, here please clarify “prophylactic administration” and “therapeutic administration”. How the doses of prophylactic and therapeutic administration were decided. Information of “prophylactic administration” and “therapeutic administration” must be included.
Response 11. The experimental steps of "prophylactic administration" and "therapeutic administration" have been described in detail in 4.6.
Point 12. Please provide data for induction of PF using different concentration TGF-β1
Response 12. Line 295 provides a reference for 10ng/mL TGF-β1 induction of PF. Because TGF-β1 induction concentration has been used in many published articles, direct reference was made to the reported concentration of action in the literature. Moreover, from the data of different indicators of TGF-β1 group and blank control group, it can be seen that the method of establishing pulmonary fibrosis cell model is feasible.
Point 13. Results of Fig 2B must be elaborated so as to reach appropriate conclusion.
Response 13. A note has been added to the results
Point 14. Results of Fig 3 needed to be analyzed and written elaborately followed by conclusion and discussion of the results.
For ex: Fig 3B, Vimentin and MMP2 expression is higher than TGF-β1 (10ng/ml) at TGF-β1 (10ng/ml) + CEP (0.2ug/ml). Likewise other results must be analyzed and incorporated appropriately.
Response 14. In Figure 3, although the expression of Vimentin and MMP2 was higher than that of TGF-β1 (10ng/ml) + CEP (0.2ug/ml), the results showed no statistical difference. Therefore, the effect of 0.2ug/ml CEP on the expression of Vimentin and MMP2 in TGF-β1 (10ng/ml) -induced MRC-5 cells was not significant.
Methods:
Point 15. There is no reference for the method used to do the said study. Please incorporate references for PF model generation, H&E staining, Masson’s trichrome staining, Cell culture and transfection, cell viability determination and apoptosis, Wound-healing assay, RNA extraction and RT–qPCR, Western blotting etc.
Response 15. A reference to the PF model building method is added at line 252. H&E staining, Masson tricolor staining, cell viability assay and apoptosis, wound healing assay, RNA extraction and RT-qPCR, and western blot are relatively common experimental techniques and therefore do not need reference
Point 16. Animal Ethics approval reference number must be provided.
Response 16. The Animal Ethics Approval Reference number is provided in line 268.
Point 17. How author selected two different CEP concentrations 5 and 15 mg/kg, and PFD (100 mg/kg) to treat BLM-induced rat. How specifically these doses were selected? What are the criteria to select treatment time and dose of CEP and PFD and in which solvent CEP was dissolved. Please incorporate information along with reference.
Response 17. CEP concentrations of 5 and 15 mg/kg were explored in a previous experiment, dissolved in an acetic acid solution with a PH of 3.7, for reference to our published article: DOI: 10.3390/ moleces27092745. PFD (100 mg/kg) was dissolved in 0.5% sodium carboxymethyl cellulose to prepare a suspension. References have been added in line 254. The duration of CEP treatment is 21 days, which is related to the time of BLM-induced establishment of pulmonary fibrosis rat model. The successful establishment of BLM-induced pulmonary fibrosis rat requires 3-4 weeks, which has been provided as a reference in line 252.
Point 18. Line 259, it is written “For cell treatment, the cells were divided into five groups: control group, TGF-β1 induced PF group, and TGF-β1-induced PF with 0.2, 0.4, 0.8, or 1 μg/mL CEP groups.” Here how do the author came to this concentration of CEP? Please clarify along with references
Response 18. In line 131, the results in Figure 2A showed that 0.2μg/mL of CEP inhibited myofibroblast proliferation during prophylactic administration, while the survival rate of myofibroblasts higher than 2μg/mL of CEP remained constant and did not change, which may be the peak of CEP inhibition. Therefore, 0.2, 0.4, 0.8 or 1μg/mL CEP was selected.
Point 19. Why cells were induced with 10 ng/mL of TGF-β1? Mention reference.
Response 19. The reference to TGF-β1 (10 ng/mL) induced cells was added in 4.5.
Point 20. Why cells were pretreated with TGF-β1? And why CEP treatment was given for 1 hour while with TGF-β1 cells were incubated for 48 hours. Till this time CEP effect might be nullified? Please Justify.
Response 20. As detailed in 4.6, in prophylactic administration, different concentrations of CEP were first treated for 1 hour, then 10ng/mL of TGF-β1 was added to the blank group for co-incubation for 48h, and CEP was not removed. In the therapeutic administration, TGF-β1 was not removed when 10ng/mL of TGF-β1 was added to the blank group for 1h, and then CEP of different concentrations was added to co-incubate for 48h.
Point 21. Also, how BLM concentration decided to be 5mg/kg to induce PF.
Response 21. In line 252, a reference has been added.
Point 22. How many animals were included in each group of animals. Author made 5 groups of animals but standard drug control group was not included. What about the vehicle and sham control group (not sprayed in the trachea by saline). It must have been included for appropriate conclusion of results. Author included one animal group to which PFD treatment was given. What is the importance of PFD. Why PFD treatment was given along with CEP. Also, why PFD and CEP treatment concentration was different. Please clarify and incorporate analysis and justify along with reference.
Response 22. Line 234. There are six animals in each group.
Line 237, the blank control group was given endotracheal atomization with normal saline. PFD is a drug approved by FBA for the treatment of pulmonary fibrosis. As a positive drug, it can be compared with the efficacy of CEP.
The concentration of CEP was explored in our previous study. Relevant literature has been published. For details, please refer to DOI: 10.3390/ moleces27092745.
The treatment concentrations of PFDS are listed in line 254.
Point 23. Why specifically two markers α-SMA and COL1A1 were considered to be studied in this study. Please provide importance of these two markers (α-SMA and COL1A1) in the present study.
Response 23. The relevant content has been added to line 65.
Point 24. Why different staining like H&E, Masson’s trichrome staining was carried out. What is its importance. Please provide information along with reference.
Response 24. H&E staining was used to observe the inflammatory manifestations and Masson staining was used to observe the hyperplasia of blue collagen fibers, which are common techniques.
Point 25. In 4.6, author mentioned in line 272 “…basal medium mixed with CCK8 dye……”. Please provide details of basal medium.
Response 25. The composition of MEM basal medium is described in 4.5.
Point 26. Results of Fig 4(A-D) needed to be described carefully and elaborately.
Response 26. Representations of the results in Figure 4 (A-D) have been added
For western it is written in line 150 “Western blotting revealed that p-p65/p65 and NLRP3 were upregulated by TGF-β1 while IκBα was downregulated”. However, careful analysis shows, p-p65/p65 expression little higher only at CEP conc 0.4 ug/ml (Fig 4c) than control whereas in other conc of CEP, the expression level is mostly equivalent to control. Similarly NLRP3 shows upregulated expression than control but at CEP conc 0.2 ug/ml NLRP3 shows more expression TGF-β1 (Fig 4D). Also IκBα expression was increased with increasing conc of CEP (Fig 4A).
Both blank control group and different CEP concentration groups were compared with TGF-β1 group to show the inhibitory effect of CEP on the related indicators of pulmonary fibrosis model cells.
Similarly results of Fig 3 also must be written elaborately. In Fig 3B, protein expression of vimentine and MMP-2 is little higher than TGF-β1at CEP conc 0.2 ug/ml. Results must be written attentively with appropriate justification.
The expression of Vmentin and MMP-2 in Figure 3 showed that the CEP concentration of 0.2ug/ml was slightly higher than that of TGF-β1, but the difference between the two groups was not statistically significant, so it could be considered that the effects of the CEP concentration of 0.2ug/ml and TGF-β1 were comparable
Point 27. Raw data of all western blot must be provided.
Response 27. Original blotting results have been provided at the time of the initial submission.
Figs/tables
Point 28. Please incorporate sufficient details of all the Figs in the figure legend.
Response 28. The description in the diagram notes has been added.
Point 29. Full form of all abbreviation must be incorporated as a foot note.
Response 29. There are not many abbreviations in the article, and most of them are common abbreviations, and the full name and abbreviation are indicated when they first appear. So I don't think it's necessary to list all the abbreviations individually.
Point 30. Alveolar septa, alveolar collapse, alveolar cavity and high granulocyte infiltration in H&E staining in Fig 1B must be labelled. Similarly, proliferative deposition must be labelled in masson staining.
Response 30. Indicated by arrows.
Point 31. In Fig 1E, BLM concentration (5mg/kg) is not written in densitometric analysis. Please incorporate.
Response 31. Changed.
Point 32. In wound healing assay photomicrographs were captured by using which instrument. Also in 4.6 author used which company ultraviolet spectrophotometer. Please add information
Response 32. Related information has been added.
Discussion:
Point 33. Discussion needed to be improved. It is not sufficiently strong.
Response 33. The presentation of the discussion has been improved.
Conclusion:
Point 34. CEP is already known as anti-inflammatory agent. In this study the obtained results were not analyzed appropriately. Some results were written very short thus led to inappropriate conclusion. The mechanism that CEP alleviates the occurrence and progression of PF is not sufficiently justified. Also conclusion portion is missing. The authors did not optimized the CEP concentration and thus the best CEP concentration was not reported.
Response 34. The presentation and analysis of results and conclusions have been increased. The results of prophylactic administration in Figure 2A showed that 0.2μg/mL-1μg/mL was the appropriate concentration for CEP.
Others
Point 35. Line 47, write correctly “abovementioned”
Response 35. Changed.
Point 36. Line 180, Correct the sentence “Consistently, We observed that TGF-β1–induced FMT which myofibroblast differentiation, proliferation, and migration”.
Response 36. Changed.
Point 37. Please incorporate full form of all abbreviation (FITC/PI, NMRC, PBS, RIPA, BCA, ECL, CST, ERK, JNK, NLPP3 etc). Full form of all abbreviation must appear when appear 1st time in the manuscript.
Response 37. PBS, RIPA and BCA are commonly used biological reagents, and people are generally more familiar with their abbreviations. AnnexinV-FITC/PI double staining is a commonly used assay for apoptosis. The rest have been added to complete statements.

Round 2
Reviewer 4 Report
Title: Cepharanthine ameliorates pulmonary fibrosis by inhibiting 2 the NF-κB/NLRP3 pathway, fibroblast-to-myofibroblast transi- 3 tion and inflammation
Revised Comments
Response to most of the comments are not satisfactory. Relevant information, references are missing and also the result analysis has not been written appropriately
1) Response to point 9 ie regarding western analysis is not justified. Western analysis is most appropriate to validate the expression results.
2) Point 10, Figure 3 is having number of subfigures. Author must provide information of relevant figure appropriately showing transformation marked by high α-SMA expression
3) Point 11, author did not provide satisfactory clarification in 4.6
4) In response to Point 13, it is written, “A note has been added to the results”, but where is this note. Author did not incorporated satisfactory analysis
5) Response to Point 15 is not satisfactory.
6) Response to Point 15, 20 is unsatisfactory
7) Response to point 22 is not matching in the manuscript
8) Response to 23, line 65 is not appropriate
9) Response 24 is not incorporated
10) Response to 25, there is no additional information in the manuscript. Its not clear
11) Response to 26 is not satisfactory. The information provided in line 150 is not matching.
12) As per response 28, sufficient information in figure legend has not been added
In conclusion, there is no clarity in result analysis and discussion.

Author Response
Response to Reviewer 4 Comments
- Response to point 9 isregarding western analysis is not justified. Western analysis is most appropriate to validate the expression results.
Response:Appropriate descriptions have been added to 2.1.
- Point 10, Figure 3 is having number of subfigures. Author must provide information of relevant figure appropriately showingtransformation marked by high α-SMA expression
Response:"the transformation is marked by high α-SMA expression" was deleted. In Figure 3, both WB and RT-qPCR results of 10ng/mL TGF-β1 group showed MRC-5 cells exposed to 10ng/mL TGF-β1 expressed higher myofibroblast marker levels, including fibronectin, COL1A1, vimentin, α-SMA, and Matrix Metallopeptidase 2 (MMP2)
- Point 11, author did not provide satisfactory clarification in 4.6
Response:"prophylactic administration" is to add different concentrations (0, 0.1, 0.2, 0.4, 0.8, 1, 2, 4, 8, and 16μg/mL) of CEP first, without removing CEP, then add 10ng/mL TGF-β1 1h later, and continue the culture 48h.
"therapeutic administration" is to add 10ng/mL TGF-β1 for 1h, without removing TGF-β1 for 1h and then add different concentrations (0, 0.1, 0.2, 0.4, 0.8, 1, 2, 4, 8, and 16μg/mL) of CEP, and continue the culture 48h.
- In response to Point 13, it is written, “A note has been added to the results”, but where is this note. Author did not incorporated satisfactory analysis
Response:Appropriate descriptions have been added to 2.2.
- Response to Point 15is not satisfactory.
Response:References to relevant experimental techniques have been added.
- Response toPoint 15, 20 is unsatisfactory
Why cells were pretreated with TGF-β1?
Response: TGF-β1 preconditioning was designed to explore the phenomenon of therapeutic administration.
And why CEP treatment was given for 1 hour while with TGF-β1 cells were incubated for 48 hours. Till this time CEP effect might be nullified? Please Justify.
Response: TGF-β1 was administered one hour after CEP treatment in order to explore the preventive effect of CEP. TGF-β1 (10 ng/mL)-induced PF with 0.2, 0.4, 0.8, or 1 μg/mL CEP groups was first treated with different concentrations of CEP for 1h. Then TGF-β1 (10 ng/mL)-induced PF group and TGF-β1 (10 ng/mL)-induced PF with 0.2, 0.4, 0.8, or 1 μg/mL CEP groups were treated with 10 ng/mL TGF-β1 for 48h, because it takes 48h for 10 ng/mL TGF-β1 to induce MRC-5 cells to become myofibroblasts. It is meaningful to administer CEP for 1h in advance. The specific results are shown in Figure 2A. Prophylactic administration of CEP has more obvious inhibitory effect on myofibroblasts than therapeutic administration of CEP
- Response to point 22 is not matching in the manuscript
1.How many animals were included in each group of animals.
Response: This is described in 4.2 “each group had six rats”
2.Author made 5 groups of animals but standard drug control group was not included. What about the vehicle and sham control group (not sprayed in the trachea by saline). It must have been included for appropriate conclusion of results.
Response: This grouping design is something we've explored before. For details, please refer to DOI: 10.3390/ moleces27092745. Don't need a standard drug control group to make that clear.
Why PFD treatment was given along with CEP. Also, why PFD and CEP treatment concentration was different. Please clarify and incorporate analysis and justify along with reference.
Response: As a positive control drug, PFD should be treated for the same duration as CEP, so PFD and CEP are simultaneously interventional. PFD has been studied a lot. The dosage of PFD has been referred to in 4.2, while the dosage of CEP has been explored by us. For specific details, please refer to our previous published articles: DOI: 10.3390/ moleces27092745.
- Response to 23, line 65 is not appropriate
Response: The number of lines in revision mode may be incorrect. The importance of α-SMA and COL1A1 is described in the third paragraph of the Introduction. Therefore, the description is not repeated in 2.1.
- Response 24 is not incorporated
Response: It is described in 2.1: “Compared with the control group, hematoxylin eosin (H&E) staining in the PF rats demonstrated that the alveolar septa and alveolar collapse had thickened (Green ar-row) and H&E staining was accompanied by high granulocyte infiltration (Black ar-row). Conversely, these symptoms were significantly relieved following 15mg/kg CEP and 100mg/kg PFD treatment, while 5mg/kg CEP did not significantly improve BLM-induced lung inflammation”, instructions H&E staining was used to observe the inflammatory manifestations.
It is described in 2.1: “Masson staining demonstrated proliferative deposition of blue collagen fibers in the lung of PF rats (Black arrow), which exhibited improvement after 15mg/kg CEP and 100mg/kg PFD treatment. However, 5mg/kg CEP did not significantly improve BLM-induced lung collagen fiber generation in rats”, instructions Masson staining was used to observe the hyperplasia of blue collagen fibers.
- Response to 25, there is no additional information in the manuscript. Its not clear
Response: Details of basal medium have been added in the first paragraph of 4.6.
- Response to 26 is not satisfactory. The information provided in line 150 is not matching.
For western it is written in line 150 “Western blotting revealed that p-p65/p65 and NLRP3 were upregulated by TGF-β1 while IκBα was downregulated”. However, careful analysis shows, p-p65/p65 expression little higher only at CEP conc 0.4 ug/ml (Fig 4c) than control whereas in other conc of CEP, the expression level is mostly equivalent to control. Similarly NLRP3 shows upregulated expression than control but at CEP conc 0.2 ug/ml NLRP3 shows more expression TGF-β1 (Fig 4D). Also IκBα expression was increased with increasing conc of CEP (Fig 4A).
Response: The results of Figure 4 have been reanalyzed and expressed in 2.4.
Similarly results of Fig 3 also must be written elaborately. In Fig 3B, protein expression of vimentine and MMP-2 is little higher than TGF-β1at CEP conc 0.2 ug/ml. Results must be written attentively with appropriate justification.
Response: The results of Figure 3 have been reanalyzed and expressed in 2.3.
- As per response 28, sufficient information in figure legend has not been added
Response: The description in the diagram notes has been added.

Round 3
Reviewer 4 Report
Re-revise comments
1) Author must pay attention while writing manuscript and response to the comments. This manuscript is suffering because of insufficient information and less attention to the results generated. Still the result section and figure legend is not connected appropriately. The information of each results has not been sufficiently detailed. It is not clear from the response that whether the information is added after the comments raised or it was already existing. If the relevant information has been added after the query raised then it should be mentioned that “it is now added”.
2) The response to very 1st comment about Western blot in result section is confusion.
In the result section it is written in line 112 “Next, we measured the α-SMA and COL1A1 expression using Western blotting suggesting that 15mg/kg CEP and 100mg/kg PFD could effectively reduce the changes of protein levels while 5mg/mg CEP did not reduce the expression levels of these proteins (Figure 1D)”. Here Fig 1D is not western blot results. It has to be rectified.
Figure legend of Fig 1D, is not matching with the results mentioned in result section. There is no figure legend of Fig 1E
3) There is no details in result section about the result shown in Fig 2B and 2C. Densitometric analysis of Fig 2C could have been provided which could have led to the appropriate conclusion showing fold change.
Figure legend of Fig 2B, C is written in line 161 as “Effect of CEP on myofibroblast migration with wound-healing assay”. It is very much insufficient.
Similarly results shown in Fig 2D,E is not sufficiently detailed in result section. It is only written “apoptosis rate was proportional to the concentration of CEP while promoting their apoptosis (Figure 2D, E)”.
Also insufficient information has been provided in Figure legend of Fig 2D, E .
4) Figure legend 3 A, B, C, D, E, F is written as “Protein levels of fibronectin, COL1A1, vimentin, MMP-2, and α-SMA measured with Western blotting” which is not matching. Moreover the word “respectively” must have been added after western blotting for meaningful sentence.
Line 166, it is written “Compared to cells from a control group, MRC-5 cells … of the signaling pathway”. The line is not justified. The relevant Fig number for this line must have been provided.
5) Response to the comment No. 7(2) ie “Author made 5 groups ….appropriate conclusion of results” is not satisfactory.
To my understanding standard drug control group, vehicle and sham control group (not sprayed in the trachea by saline) should have been included in every experimental set up. This is because each time experimental set up condition vary depending upon various conditions (such animal procurement time, size, environment etc) and hence the results may not match with earlier experiments results specifically in-vivo experiments. I suggest to repeat the experiments.
6) Fig legend of “Fig 4 (A, C, D, E): Protein levels of NLRP3, phosphorylated p65(p-p65), total 202 p65, and IκBα measured with Western blotting” is not matching with the Fig 4 (A, C, D, E).

Author Response
Response to Reviewer 4 Comments
1) Author must pay attention while writing manuscript and response to the comments. This manuscript is suffering because of insufficient information and less attention to the results generated. Still the result section and figure legend is not connected appropriately. The information of each results has not been sufficiently detailed. It is not clear from the response that whether the information is added after the comments raised or it was already existing. If the relevant information has been added after the query raised then it should be mentioned that “it is now added”.
Response:In the previous rounds of revision, because the revision mode was uploaded, there was a difference in the information obtained between the author and the reviewer, which was my improper handling. The revision mode of this revised manuscript has been closed. Compared with the previous round of manuscript, the revised parts have been manually marked in red.
2) The response to very 1st comment about Western blot in result section is confusion.
In the result section it is written in line 112 “Next, we measured the α-SMA and COL1A1 expression using Western blotting suggesting that 15mg/kg CEP and 100mg/kg PFD could effectively reduce the changes of protein levels while 5mg/mg CEP did not reduce the expression levels of these proteins (Figure 1D)”. Here Fig 1D is not western blot results. It has to be rectified.
Response:In the last round of revision, figure 1 has modified, deleted the original figure 1D, and changed the original figure 1E to figure 1D.
Figure legend of Fig 1D, is not matching with the results mentioned in result section. There is no figure legend of Fig 1E
Response:In the last round of revision, figure 1 has modified, deleted the original figure 1D, and changed the original figure 1E to figure 1D.
3) There is no details in result section about the result shown in Fig 2B and 2C. Densitometric analysis of Fig 2C could have been provided which could have led to the appropriate conclusion showing fold change.
Response:it is now added and the changed part has been highlighted in red.
Figure legend of Fig 2B, C is written in line 161 as “Effect of CEP on myofibroblast migration with wound-healing assay”. It is very much insufficient.
Response:it is now added and the changed part has been highlighted in red.
Similarly results shown in Fig 2D,E is not sufficiently detailed in result section. It is only written “apoptosis rate was proportional to the concentration of CEP while promoting their apoptosis (Figure 2D, E)”.
Response:it is now added and the changed part has been highlighted in red.
Also insufficient information has been provided in Figure legend of Fig 2D, E .
Response:it is now added and the changed part has been highlighted in red.
4) Figure legend 3 A, B, C, D, E, F is written as “Protein levels of fibronectin, COL1A1, vimentin, MMP-2, and α-SMA measured with Western blotting” which is not matching. Moreover the word “respectively” must have been added after western blotting for meaningful sentence.
Response:Related mismatches in the legend have been changed and marked in red. The word “respectively” has been added after western blotting.
Line 166, it is written “Compared to cells from a control group, MRC-5 cells … of the signaling pathway”. The line is not justified. The relevant Fig number for this line must have been provided.
Response:it is now added and marked in red.
5) Response to the comment No. 7(2) ie “Author made 5 groups appropriate conclusion of results” is not satisfactory. To my understanding standard drug control group, vehicle and sham control group (not sprayed in the trachea by saline) should have been included in every experimental set up. This is because each time experimental set up condition vary depending upon various conditions (such animal procurement time, size, environment etc) and hence the results may not match with earlier experiments results specifically in-vivo experiments. I suggest to repeat the experiments.
Response:I think we should follow the 'single variable principle' when setting groups. The blank control group was injected with normal saline, and the other groups were injected with the same amount of BLM, which followed the principle of "single variable". In addition, I also referred to a lot of published literature (as described below), none of them set sham control group (not sprayed in the trachea by saline), so I think my grouping is reasonable and there is no need to set a sham control group (not sprayed in the trachea by saline).
Reference:
- Peng L, Wen L, Shi QF, Gao F, Huang B, Meng J, Hu CP, Wang CM. Scutellarin ameliorates pulmonary fibrosis through inhibiting NF-κB/NLRP3-mediated epithelial-mesenchymal transition and inflammation. Cell Death Dis. 2020 Nov 13;11(11):978. doi: 10.1038/s41419-020-03178-2. PMID: 33188176; PMCID: PMC7666141.
- Yao H, Wei S, Xiang Y, Wu Z, Liu W, Wang B, Li X, Xu H, Zhao J, Gao Y. Kangfuxin Oral Liquid Attenuates Bleomycin-Induced Pulmonary Fibrosis via the TGF-β1/Smad Pathway. Evid Based Complement Alternat Med. 2019 Nov 3;2019:5124026. doi: 10.1155/2019/5124026. PMID: 31885648; PMCID: PMC6926420.
- Pei X, Zheng F, Li Y, Lin Z, Han X, Feng Y, Tian Z, Ren D, Cao K, Li C. Niclosamide Ethanolamine Salt Alleviates Idiopathic Pulmonary Fibrosis by Modulating the PI3K-mTORC1 Pathway. Cells. 2022 Jan 20;11(3):346. doi: 10.3390/cells11030346. PMID: 35159160; PMCID: PMC8834116.
6) Fig legend of “Fig 4 (A, C, D, E): Protein levels of NLRP3, phosphorylated p65(p-p65), total 202 p65, and IκBα measured with Western blotting” is not matching with the Fig 4 (A, C, D, E).
Response:Related mismatches in the legend have been changed and marked in red.
